# Work Ability and Job Survival: Four-Year Follow-Up

**DOI:** 10.3390/ijerph16173143

**Published:** 2019-08-28

**Authors:** Maria Carmen Martinez, Frida Marina Fischer

**Affiliations:** 1WAF Informatics and Health, São Paulo 04109-100, Brazil; 2Department of Environmental Health, School of Public Health, University of São Paulo, São Paulo 01246-904, Brazil

**Keywords:** work ability, life course, aging, longitudinal studies, prolonged work career, healthcare worker

## Abstract

Background: Employees with impaired work ability might be at higher risk of remaining shorter in the job than those with adequate work ability. The aim of the study was to establish whether work ability plays a role in job survival. Methods: Four-year follow-up (2008–2012) study of 1037 employees of a hospital in São Paulo, Brazil. Work ability was categorized as “adequate” or “impaired”. Employment status at the end of follow-up was categorized as active, resignation or dismissal. Survival analysis was performed using the Kaplan–Meier method and the Cox proportional-hazards model. Results: About 78.9% of the participants had adequate and 21.1% impaired work ability. Job survival was longer for the participants with adequate work ability independently from the type of job termination (*p* < 0.001). The odds of job termination were higher for the participants with impaired work ability (*p* < 0.001) who either resigned (hazard ratio—HR = 1.58) or were dismissed (HR = 1.68). Conclusion: Job survival was shorter for the employees with impaired work ability independently from the type of job termination. It was also shorter for the employees who were dismissed compared to those who resigned. Duration in the job might be extended through actions to enhance work ability.

## 1. Introduction

The most widely accepted concept of work ability is represented by the answer to the question “how good is the worker at present, in the near future, and how able is he or she to do his or her work with respect to the work demands, health and mental resources?” [1].

Impairments of the ability of workers to perform their tasks have negative direct or indirect impacts on themselves and society at large. The predictive value of work ability for several negative outcomes is well known, including physical and mental diseases, sick leave, job dissatisfaction, loss of productivity, reduced employability, unemployment, leaving the profession, early retirement, and even death [1,2,3,4,5,6,7,8]. Work ability further influences aspects such as job security, employment severance, disability retirement, return to work, relocation, precarious work, and career opportunities [4,6,9].

Work ability has multicausal determinants derived from the personal characteristics of workers, family and social factors, working conditions, and the organization of work [1,3,10,11]. Occupations and tasks characterized by high physical and mental load are associated with higher risk of impaired work ability [1,5,12]. Within this context, healthcare providers, especially those in the hospital setting, deserve special attention, because they are exposed to a large number of physical and mental stressors, such as inadequate equipment and physical space, biological hazards, responsibility for human lives, close contact with patients’ pain and suffering, low salary, low recognition, and, more recently, new and complex technologies and increasing demands for high-quality and safe care [12,13,14,15,16].

This situation is particularly worrisome in the present time, since organizations (including hospitals) are restructuring their work processes and reducing their staff [6,12,17]. In addition, several countries, including Brazil, are making thorough social security and labor reforms to reduce unemployment, control the pertinence and duration of leaves, ensure the survival of the social security system, mitigate the impact of population aging, and change the nature of work [18,19,20]. However, these reforms are attended by some undesirable effects, such as precarious labor relations, pay cuts, increase of informal work, and job insecurity [18,20].

The hypothesis underlying the present study is that employees with poorer work ability might be at higher risk of job instability. Work ability might also determine differences in job survival between employees who resign and those dismissed.

Although adequate work ability is an essential condition for workers to remain in their job, this relationship is scarcely addressed in the literature. Therefore, the aim of the present study was to establish whether work ability played a determinant role in job survival among employees of a hospital in São Paulo, Brazil, who eventually resigned or were dismissed, along 4 years.

## 2. Materials and Methods

### 2.1. Population and Study Design

The present is part of a 4-year cohort (2008–2012) study performed at a medium-sized, high-complexity private hospital in São Paulo, Brazil. Participants were 1037 out of 1212 eligible employees.

Participants did not exhibit significant difference (*p* > 0.050) in age and job tenure compared to nonparticipants. Losses among men were higher compared to those among women (18.0% vs. 10.0%; *p* < 0.001). Statistically significant difference (*p* < 0.001) was also detected for the following variables: hospital department, hospital area and position, with wide variation among the various occupational categories. Details of the studied population (with the same sample and follow-up) were published previously [9]. At baseline, the participants responded a questionnaire for demographic (sex, age, marital status, educational level), lifestyle (smoking, drinking, practice of physical activity, nutritional status), occupational variables (age at onset of work, years of work in the current profession, years of work at the institution, second job, work shift, night shift at the investigated institution or elsewhere, total weekly working time-at the job and at home, department, work area and position, psychosocial work environment), and work ability. Information on the employees’ status (active, resignation or dismissal) was obtained from the human resources department and the head of each area.

### 2.2. Measurements

Employment status at the end of follow-up (2012) was categorized as active, resignation or dismissal.

Work ability was assessed by means of the Work Ability Index–WAI, validated for use in Brazil [11,21]. WAI comprises 7 dimensions: current work ability compared to the lifetime best, work ability in relation to the job demands, number of current diseases self-reported and diagnosed by a physician, estimated work impairment due to diseases, sick leaves, own prognosis of work ability, and mental resources [11]. The total score ranges from 7 to 49, and the higher the score, the better the work ability [11]. The results were categorized as excellent, good, moderate or poor work ability, according to the criteria formulated by Kujala et al. (2005) [22] for individuals under 35 and by Tuomi et al. (2005) [11] for older workers. Work ability was dichotomized as adequate (excellent/good) or impaired (moderate/poor). The reliability of WAI was satisfactory (Cronbach’s alpha = 0.70). Appendix A presents additional information about WAI questionnaire.

Psychosocial work environment was assessed by means of the Job Stress Scale (JSS), validated for use in Brazil [23]. It is an abridged version of the Job Content Questionnaire, based on the Demand Control Model [24,25]. JSS comprises 3 scales: demands (score 5 to 20), control (score 6 to 24), and social support at work (score 6 to 24). These three variables were dichotomized as high or low exposure, using the midpoint of each scale as the cutoff point. Next, the variable “psychosocial work environment” was created and categorized as low-strain (high control/low demand), active (high control/high demand), passive (low control/low demand), and high-strain (highest risk situation, low control/high demand) jobs. This variable was dichotomized as low/moderate (low-strain + active + passive jobs) and high-strain jobs. Social support at work was dichotomized as high (better situation) and low (worse situation). JSS showed reasonable and satisfactory reliability: demands, α = 0.63; control, α = 0.81; and social support, α = 0.67.

### 2.3. Statistical Analysis

Descriptive analysis included calculation of mean, standard deviation (SD), median, maximum, and minimum values for continuous variables, and absolute and relative frequencies for categorical variables.

Survival analysis was performed with the Kaplan–Meier method to estimate the probability of surviving in each time interval. The log-rank test was used to compare accumulated survival curves between the categories of variables. Survival time was defined as the time (months) from the date of the initial assessment of work ability (2008) until failure (job termination) or the end of follow-up (2012). Risk for job termination was analyzed by means of the Cox proportional-hazards model; risk was measured as the hazard ratio (HR). Variables with *p* value < 0.200 on the log-rank test were included in stepwise multiple Cox analysis. The proportional-hazards assumption was verified through log-log plots for each variable and through Schoenfeld’s test for the final model. The model fit was evaluated by means of the likelihood ratio test. The significance level was set to *p* <0.050 in all the analyses [26,27].

### 2.4. Ethical Issues

The study was approved by the Ethics Committee of School of Public Health, University of São Paulo (ruling n^o^. 257,518) and complied with the principles in the Declaration of Helsinki and recommended by the World Medical Association. Participation was voluntary; all the participants signed an informed consent form, and the confidentiality of the results was assured.

## 3. Results

### 3.1. Descriptive Analysis

The participants’ mean age was 35.1 years old (SD = 8.4), with 29.3% over 40; 69.3% were females. Most participants were allocated to the clinical department (61.9%) and nursing services (51.7%). The largest proportions corresponded to nursing technicians (22.1%), attendants (18.9%), registered nurses (15.1%)—allocated to managerial tasks or direct patient care—and nursing assistants (14.1%).

The average score on WAI was 42.3 (SD = 4.7). Work ability was rated excellent for 418 (40.3%) participants, good for 400 (38.6%), moderate for 166 (16.0%), and poor for 53 (5.1%). Therefore, 21.1% of the sample exhibited impaired work ability. Appendix A presents a table with the results of the Work Ability Index dimensions from the studied population.

As to the outcome, 536 (51.7) participants were still active at the end of the follow-up, 148 (14.3%) had resigned. and 353 (34.0%) had been dismissed. Thus, 501 (48.3%) participants were no longer working at the institution (voluntary or involuntary employment termination) at the end of follow-up, with an annual termination rate of 12.0%. Details of the participants’ demographics, lifestyle, occupation, and exposure to occupational work stressors have been published previously [9].

### 3.2. Survival Analysis

From the group of employees no longer working at the institution (48.3%), job survival was up to 1 year for 63.5%, up to 2 years for 45.2%, up to 3 years for 32.2%, and up to 4 years for 7.2% (Table 1, Figure 1).

Job survival was higher for the participants with adequate compared to those with impaired work ability at baseline (*p* < 0.001)—64.3% and 48.4% up to 12 months; 42.4% and 24.2% up to 24 months; 24.9% and 10.9% up to 36 months; and 8.0% and 0.0% up to 48 months, respectively. The cumulative proportion of job termination for employees with adequate and impaired work ability was, respectively, 25% up to 6.9 and 4.1 months, 50% up to 19.0 and 11.0 months, and 75% up to 35.6 and 23.3 months (Table 1, Figure 2).

Job survival was longer for the employees who resigned compared to those who were dismissed (*p* = 0.022)—62.2% vs. 59.5% up to 1 year, 43.2% vs. 35.1% up to 2 years, 28.4% vs. 18.1% up to 3 years, and 7.4% vs. 5.1% up to 4 years, respectively. The cumulative proportion of job termination for employees who resigned and were dismissed was, respectively, 25% up to 6.1 and 6.7 months, 50% up to 19.1 and 17.2 months, 75% up to 39.3 and 31.1 months, and 95% in up to 49.5 and 47.7 months (Table 1, Figure 3).

Considering both variables together (work ability and job termination), survival was longer for the employees with adequate work ability, both those who resigned and those who were dismissed. For the employees with adequate work ability, the 4-year job survival rate was 10.8% for those who resigned and 7.0% for the ones who were dismissed. The 4-year job survival rate was 0.0% for all the employees with impaired work ability independently from the type of job termination (Table 1, Figure 4).

In regard to the time of job termination for the employees who resigned, 75.0% of job terminations occurred in up to 40.5 months for those with adequate work ability and in up to 30.1 months for those with impaired work ability. The corresponding times for the dismissed employees were up to 33.6 and 21.2 months for those with adequate and impaired work ability, respectively (Figure 4).

According to the Cox proportional-hazards model, the risk of job termination at the end of the follow-up was higher for workers with impaired work ability (*p* < 0.001) who either resigned (HRa = 1.58) or were dismissed (HRa = 1.68) (Table 2).

Table 2 further shows that among the employees who resigned, the risk of job termination was higher for those with overweight (HRa = 1.54; *p* = 0.024). Among the dismissed employees, the risk of job termination was higher for those allocated to the clinical/general operations department (HRa = 1.49; *p* = 0.024). The risk of job termination was higher among employees with jobs requiring higher (registered nurses), medium (nursing technicians and general technicians) or lower (waitresses) professional training level (HRa = 1.62; *p* = 0.024) or in jobs characterized as requiring low professional/technical qualification (nursing assistants, attendants, and hygiene assistants) (HR = 1.81; *p* = 0.004) compared to administrative employees.

All the variables which remained in the final model exhibited a sufficient number of events in each category, and none violated the hazard proportionality assumption. The results of Schoenfeld’s test showed that hazards were proportional in both models (*p* > 0.050). The likelihood ratio test evidenced an adequate fit (*p* ≤ 0.050).

## 4. Discussion

Our results show that the overall job survival was short; only 7.2% of the total number of employees remained 4 years in the job. The employees with adequate work ability at baseline remained longer in the job compared to those with impaired work ability, independently from the type of job termination. The results further show that the dismissed employees remained shorter in the job than those who resigned.

We could not locate any other study that analyzed work ability and job survival; therefore, we have no grounds for comparisons. Nevertheless, the high rates of short job duration we found are compatible with the known high rates of employee turnover and early exit from the profession among hospital workers. This is true particularly for nursing professionals, who represented the largest proportion of participants in the present study [9,16,28]. High turnover rates are related to the high physical and mental load of hospital work, which is characterized by daily exposure to suffering and death, shift work, long working hours, high biomechanical and cognitive load, conflicting, even violent labor relations, role conflict, low recognition and high levels of responsibility, in addition to personal reasons, such as family care [1,9,12,15,16].

According to a report recently published in the United States, workers who resigned accounted for 92.7% of all hospital job terminations [16]. In our study, of the 501 cases of job termination, 29.5% corresponded to employee resignation and 70.5% to dismissals. One possible reason to account for this discrepancy is that major changes were made at the analyzed hospital along the study period, including introduction of new and complex technologies, higher quality and safety demands, new guidelines, and redefinition of the organizational structure. Changes in care delivery, staff size and composition, demands for higher productivity and profitability, and new and higher-level responsibility and roles might elicit feelings of uncertainty and dissatisfaction and increase the workload, resulting in voluntary or involuntary termination from the job [9,12,16,29]. It should be observed that the rate of resignations decreased in Brazilian hospitals as a function of the overall slowdown of the labor market in the country [28].

As was mentioned above, work ability alludes to the workers’ perception of their physical, mental, and social resources to meet the physical and mental demands of work [1,11]. Its predictive value for several negative health outcomes, employability, and employment has already been demonstrated [1,2,3,4,5,6,7,8,9]. In the present study, work ability was the main determinant of job survival independently from the type of job termination. This finding corroborates the notion that adequate work ability is required for employability. Workers with better work ability are healthier, have more coping resources, are more productive, result in lower healthcare costs, and have less absenteeism; therefore, they have better employability [1,3,7,9,15].

Job termination might be involuntary, affecting workers who are considered undesirable due to poor skills, performance, productivity, health or patterns of behavior, high payroll impact, and/or older age [6,9,30,31]. These characteristics might be associated with impaired work ability and, consequently, with the shorter job survival found in the present study.

In turn, workers with better work ability remain longer in the job and tend only to leave when they see and are eager for new work opportunities. Voluntary termination might be motivated by a desire for better working conditions, career opportunities, less conflicting interpersonal relationships, more recognition, learning and growth opportunities, and better conditions for work adjustment to functional and/or health limitations [2,6,9,32].

In another analysis of this same population, we found that impaired work ability was a risk factor for type of job termination (namely, for dismissal but not for resignation), which indicates that workers with poorer work ability are less fit to meet job demands and labor market requirements and thus have less employability [9]. In the present study, using data of the follow-up of the same sample, work ability had an impact on time of both types of job termination.

As was shown, job survival was also influenced by other factors (overweight, department, and position). Among the workers who resigned, the odds of job termination were higher for those with overweight, but not with obesity, compared to those with normal weight. Obesity is associated with poorer performance, impaired health, and low self-confidence for job search [9,33,34]. This might, at least partially, explain why obesity was not associated with resignation.

Among the dismissed employees, the risk of job termination was higher among those allocated to the clinical/general operations department and jobs other than specialized administrative positions. Workers at higher risk for employment termination had jobs characterized by medium-level leadership or were operational staff engaged in direct patient care or support activities. These groups are often subjected to poor working conditions, including high workload, high physical and mental load, daily exposure to biological, chemical, and physical hazards, low salary, and low recognition. All these factors might cause illnesses, frequent injuries, and exhaustion, with consequent impairment of work ability [2,9,35,36]. In addition, workers with fewer skills and a lower salary are, as a rule, easy to replace, given the large supply of manpower available in the labor market [37].

Among the strengths of the present study is that, to the best of our knowledge, it is the first that analyzed job survival according to work ability and type of job termination. In addition, its longitudinal design allowed establishing some causal relationships between job survival and independent variables.

In regard to the study limitations, we cannot rule out the healthy worker effect, resulting in longer job duration for the healthier employees [6,38]. If this was the case, the rate of employees with impaired work ability and shorter job survival might have been underestimated.

Moreover, work ability was assessed at the onset of follow-up (2008) instead of at the time of hiring. This situation characterizes left censoring, i.e., participants began to be observed at a definite time, the milestone of interest having occurred previously, with its exact time unknown [26,27]. As we could not establish the participants’ previous work ability profile, we sought to control previous exposure through proxy variables such as chronological age, age at onset of working life, years in the occupation, and job tenure. None of these variables exhibited a statistically significant relationship with the outcome, which suggests that impaired work ability, even if recent, is more relevant for termination from employment than past exposure to occupational hazards and other factors.

Finally, the present study was restricted to hospital workers. Future studies should analyze a broader range of occupations and also interventions to extend job survival.

The results of the present study corroborate the notion that enhancing work ability has implications for collective policies as a function of its determinant role for job termination. High turnover and job termination have negative consequences, implicating hiring and relocation and, therefore, additional investment in selection, training, and qualification of workers [9,16,28,39]. Losing the more experienced employees and unstable staff composition contribute to reducing productivity, job dissatisfaction, stress at work, work-related diseases, and higher incidence of care-sensitive adverse events [16,39,40]. More than that, losing one’s job and leaving the workforce have implications for workers (mental health, social role, and self and family livelihood) and society at large (financial burden for the social security administration and health system) [1,2,3,16,39,40].

Recommendations for staff retention should consider how employment decisions are made, actions to build relationships, commitment, and confidence [16], and reflecting on the criteria to select the employees who will be dismissed, especially under production restructuring and downsizing conditions [9]. Since work ability is the balance between the worker’s resources and the conditions/organization at work, actions to enhance work ability should not merely seek health promotion and to prevent diseases and injuries but also and foremost to improve the physical and psychosocial work environment [1,2,6,9,41].

## 5. Conclusions

The results of the present study show that the participants remained in the job for a relatively short period of time. Employees with impaired work ability at baseline remained a relatively shorter time in the job in the short-to-medium run (4 years) independently from the type of job termination than those with adequate work ability. The results further indicate that survival in the job was shorter for the dismissed employees compared to those who resigned. Overweight, hospital department, and position also influenced job survival. Duration in the job might be extended through actions to enhance work ability.

## Figures and Tables

**Figure 1 ijerph-16-03143-f001:**
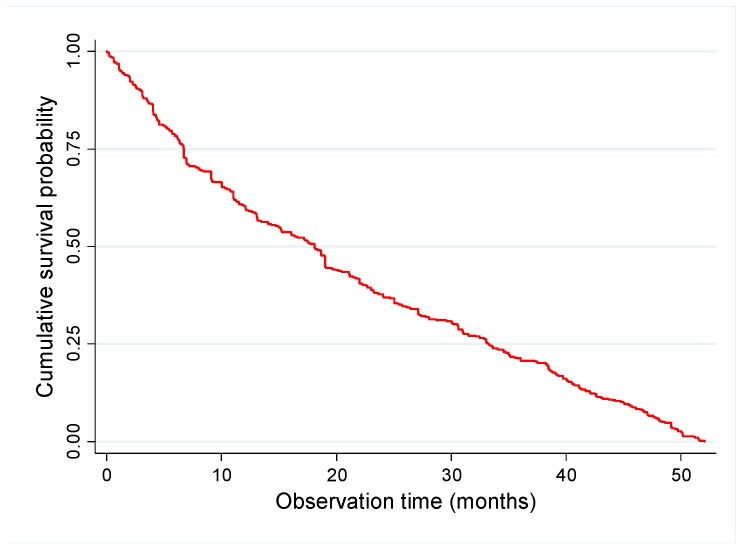
Kaplan–Meier survival curve for job termination.

**Figure 2 ijerph-16-03143-f002:**
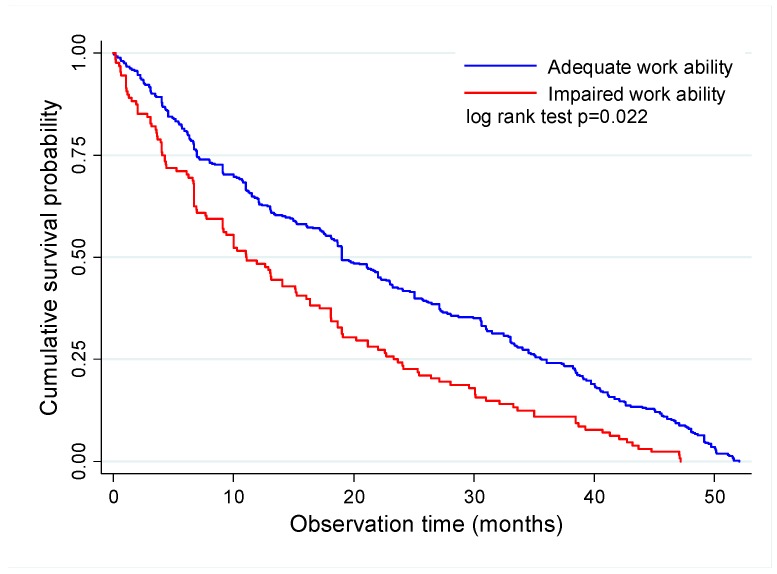
Kaplan–Meier survival curve for job termination according to work ability status.

**Figure 3 ijerph-16-03143-f003:**
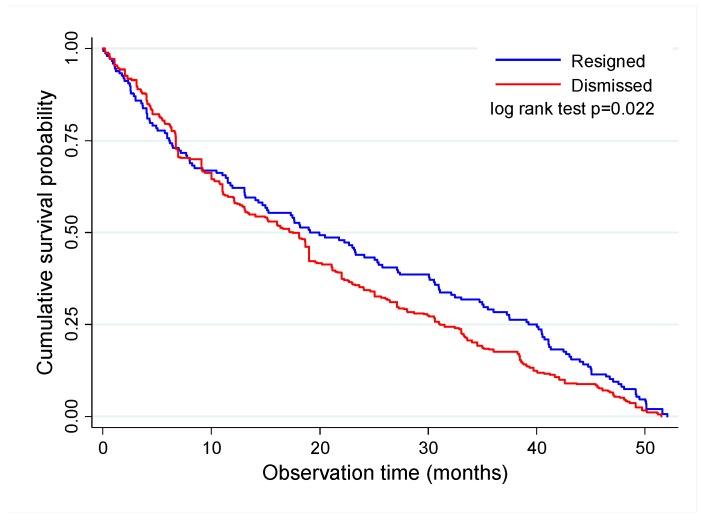
Kaplan–Meier survival curve for job termination according to type of job termination.

**Figure 4 ijerph-16-03143-f004:**
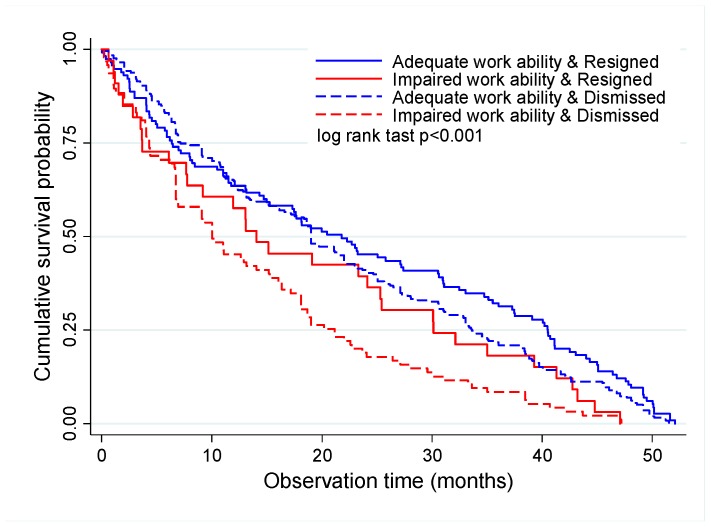
Kaplan–Meier survival curve for job termination according to work ability status and type of job termination.

**Table 1 ijerph-16-03143-t001:** Survival table according to work ability status and type of job termination.

Time (Months)	Cumulative Proportion Surviving at the Time								
	Total	Work Ability		Job Termination		Work Ability and Job Termination			
		Adequate	Impaired	Resigned	Dismissed	Adequate Work Ability—Resigned	Adequate Work Ability—Dismissed	Impaired Work Ability—Resigned	Impaired Work Ability—Dismissed
**12.0**	60.3	64.3	48.4	62.2	59.5	63.5	64.7	57.6	45.3
24.0	37.7	42.4	24.2	43.2	35.1	45.2	41.1	36.4	17.9
36.0	21.4	24.9	10.9	28.4	18.1	32.2	20.9	18.2	9.5
48.0	6.0	8.0	0.0	7.4	5.1	10.4	7.0	0.0	0.0

**Table 2 ijerph-16-03143-t002:** Results of the Cox multiple regression analysis.

Variable	Resigned*			Dismissed**		
	HRa	95% Confidence Interval	*p*-Value	HRa	95% Confidence Interval	*p*-Value
**Work ability**						
Adequate	1.00			1.00		
Impaired	1.58	1.05–2.38	0.029	1.68	1.31–2.14	<0.001
**Department**						
Others				1.00		
Clinical/General operations				1.49	1.14–1.95	0.004
**Position**						
Administrative specialist				1.00		
Registered nurse (management or patient care) Technician/Nursing technician/Waitress				1.62	1.07–2.47	0.024
Nursing assistant/Assistant or Attendant/Cleaner				1.81	1.20–2.72	0.004
**Nutritional status**						
Normal	1.00					
Overweight	1.54	1.06–2.25	0.024			
Obesity	0.61	0.24–1.56	0.300			
**Sex**						
Female	1.00					
Male	0.81	0.54–1.22	0.313	.	.	.

Note: HRa = adjusted hazard ratio; * analysis adjusted for sex; ** analysis was not adjusted for sex or age, since the variables did not exhibit proportional hazards.

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
