# Peer review of "Work Ability and Job Survival: Four-Year Follow-Up"

_ijerph, 2019, doi:10.3390/ijerph16173143_

Round 1

Reviewer 1 Report

Well designed and written research paper. My only concern is about the really high turnover rate of the hospital employees during the 4-year follow-up, which the authors also discuss. However, the results of the analysis are based on solid design, appropriate statistical methods, and in a sense predictable and thus valid.

The authors have published recently another analysis of the same population, unfortunately I only had access to the abstract of that paper ( doi.org:10.1097/JOM.0000000000001599). Based on the abstract it seemed to have a shorter follow-up and different design. Thus the results of this paper presents novel results of the subject area.

On page 5 row 202 the latter 7.0% should be removed.

Editorial minor notice: In the reference list the centered alignment of the text makes the word spaces too wide before active internet links

Author Response

Work ability and job survival: four-year follow-up

Maria Carmen Martinez, Frida Marina Fischer

Comments of the first reviewer

We thank all comments of the reviewer, and try to comply with suggestions. Replies were performed point to point. The included changes are highlighted in red color in the updated version of the manuscript.

Point 1) Well designed and written research paper. My only concern is about the really high turnover rate of the hospital employees during the 4-year follow-up, which the authors also discuss. However, the results of the analysis are based on solid design, appropriate statistical methods, and in a sense predictable and thus valid.

Reply 1: Thank you for your compliments.

Point 2) The authors have published recently another analysis of the same population, unfortunately I only had access to the abstract of that paper ( doi.org:10.1097/JOM.0000000000001599).  Based on the abstract it seemed to have a shorter follow-up and different design. Thus the results of this paper present novel results of the subject area.

Reply 2: We would like to clarify that both studies refer to the same population, and the same follow-up.  In the first study published in JOEM 2018, factors associated with the type of job termination (dismissal or resignation) were evaluated.  In this paper we evaluated the TIME until job termination (job survival).  In the study published in JOEM 2018, we showed that work ability was a risk factor for job dismissal, but not for job resignation.  This current paper (IJERPH) shows work ability impact both job dismissal and job termination. This is underlined on page 7, line 290. We have made a modification in the final sentence of the paragraph starting at line 293 to better clarify the differences between the two articles.  We also included this information on line 68.

Point 3) On page 5 row 202 the latter 7.0% should be removed

Reply 3 :  It was removed ( line 206).

Point 4) Editorial minor notice: In the reference list the centered alignment of the text makes the word spaces too wide before active internet links

Reply 4: References have been realigned on the left.

Reviewer 2 Report

This is a paper that is a 4-year follow-up study on work ability referring to a previously published study.  I commend the use of a follow-up using the same sample!  The paper aims to investigate the “job survivability” of workers as it relates to their levels of work ability.  Using a four-year follow-up study, you were able to track your sample group’s levels of work ability and relate that to their employment status over that time period.  I really liked your methodology – this was a study that went above and beyond a simple point-in-time correlation and provided comparisons of real outcomes. 

I think you have an interesting topic here that potentially has far reaching impacts on how workers are managed across industries and markets, and what factors organizational practitioners should be paying attention to in designing worker support and benefit programs.  For example, the point that nutritional status was identified as a significant factor is very interesting and very useful.  I also believe your paper provides a major contribution to our understanding of work ability and it’s impacts on workers and organizations alike.  Additionally, as you stated, this may be the first study of its kind. 

Your ultimate findings were interesting, and I think telling.  The finding that workers with impaired work ability remained in their jobs for shorter periods of times than those with adequate work ability is extremely important.  I again think that your choice of a time lagged study is a major strength of this study that nicely supports the validity of your findings.  I think it would / will be interesting to see if this type of study is replicated in other industry or market settings to see if the same results occur. 

I think there are a few minor things you could add or adjust to make this manuscript stronger. 

I am curious to know if there were other physical conditions / ailments that were not used in the model other than “nutritional status”.  Could you add either some commentary or a table to address this?

I would like to see a table outlining the descriptive demographic characteristics either in addition to, or to replace the text outline of them.  You discuss them, but the verbiage explaining them on page 3 begins to get a bit confounding and having a table to refer to, I think, would make those demographics clearer to the reader. 

I would also like to see either a table outlining all of the WAI dimensions and what your results for those were, or at least the addition of a sample of the instrument to give the reader a flavor of the types of questions respondents were being asked – did the questions provide predefined categories or were they open ended questions asking respondent for self-identified conditions?  You list them in the text on page 2, but I think that a bit more clarity on them would be useful. 

There are several typo’s and grammatical errors throughout the paper.  I won’t rehearse them in detail here but feel that a thorough proofreading is needed before a final accept.  Overall, nicely done!

Author Response

Work ability and job survival: four-year follow-up

Maria Carmen Martinez, Frida Marina Fischer

Comments of the second reviewer

We thank all comments of the reviewer, and try to comply with suggestions. Replies were performed point by point. The included changes are highlighted in red color in the updated version of the manuscript.

Point 1) This is a paper that is a 4-year follow-up study on work ability referring to a previously published study.  I commend the use of a follow-up using the same sample!  The paper aims to investigate the “job survivability” of workers as it relates to their levels of work ability.  Using a four-year follow-up study, you were able to track your sample group’s levels of work ability and relate that to their employment status over that time period.  I really liked your methodology – this was a study that went above and beyond a simple point-in-time correlation and provided comparisons of real outcomes. 

Reply 1: This is underlined on page 7, line 290. We have made a modification in the final sentence of the paragraph starting at line 293 to better clarify the differences between the two articles (JOEM, 2018, and the current manuscript). We also included additional information on line 68.

Point 2) I think you have an interesting topic here that potentially has far reaching impacts on how workers are managed across industries and markets, and what factors organizational practitioners should be paying attention to in designing worker support and benefit programs.  For example, the point that nutritional status was identified as a significant factor is very interesting and very useful.  I also believe your paper provides a major contribution to our understanding of work ability and it’s impacts on workers and organizations alike.  Additionally, as you stated, this may be the first study of its kind. 

Your ultimate findings were interesting, and I think telling.  The finding that workers with impaired work ability remained in their jobs for shorter periods of times than those with adequate work ability is extremely important.  I again think that your choice of a time lagged study is a major strength of this study that nicely supports the validity of your findings.  I think it would / will be interesting to see if this type of study is replicated in other industry or market settings to see if the same results occur. 

Reply 2: Thanks for the compliments. Recently, the Brazilian Congress passed important labor and social security reforms. We believe these changes, which restrict workers 'rights, will have more significant impacts on both work ability and workers' employment. Our expectation is to contribute beyond academic knowledge, subsidizing public policies and work changes.

Point 3) I think there are a few minor things you could add or adjust to make this manuscript stronger.  I am curious to know if there were other physical conditions / ailments that were not used in the model other than “nutritional status”.  Could you add either some commentary or a table to address this?

Reply 3: Thank you for your suggestion. We add a list of the investigated variables. Please see on line 69 to 74 : At baseline, the participants responded a questionnaire for demographic (sex, age, marital status, educational level), lifestyle (smoking, drinking, practice of physical activity, nutritional status), occupational variables (age at onset of work, years of work in the current profession, years of work at the institution, second job, work shift, night shift at the investigated institution or elsewhere, total weekly working time-at the job and at home, department, work area and position, psychosocial work environment) and work ability.

Point 4) I would like to see a table outlining the descriptive demographic characteristics either in addition to, or to replace the text outline of them.  You discuss them, but the verbiage explaining them on page 3 begins to get a bit confounding and having a table to refer to, I think, would make those demographics clearer to the reader. 

Reply 4:  The table with demographic characteristics and other variables was published in the previous article (JOEM 2018, reference 9). For this reason, we chose not to reprint it in this article. We kept only the text with the main information. If you are interested we can send a copy of the article published in JOEM for private use (does not have open-access status). Please, send us your e-mail, or ask the editor, to provide your email.

Point 5) I would also like to see either a table outlining all of the WAI dimensions and what your results for those were, or at least the addition of a sample of the instrument to give the reader a flavor of the types of questions respondents were being asked – did the questions provide predefined categories or were they open ended questions asking respondent for self-identified conditions?  You list them in the text on page 2, but I think that a bit more clarity on them would be useful. 

Reply 5:  Additional tables included in the manuscript cannot be included, as exceed the maximum number of allowed figures/tables. Per your suggestion, we prepared and included two supplementary files:  Supplementary 01, with additional information about Work Ability Index (WAI). And supplementary 02, showing results of WAI dimensions of the studied population.

Point 6) There are several typo’s and grammatical errors throughout the paper.  I won’t rehearse them in detail here but feel that a thorough proofreading is needed before a final accept.  Overall, nicely done!

Reply 6: a general revision was performed to correct grammatical and typos errors.

This manuscript is a resubmission of an earlier submission. The following is a list of the peer review reports and author responses from that submission.

Round 1

Reviewer 1 Report

Well designed and written research paper. My only concern is about the really high turnover rate of the hospital employees during the 4-year follow-up, which the authors also discuss. However, the results of the analysis are based on solid design, appropriate statistical methods, and in a sense predictable and thus valid. 

The authors have published recently another analysis of the same population, unfortunately I only had access to the abstract of that paper ( doi.org:10.1097/JOM.0000000000001599). Based on the abstract it seemed to have a shorter follow-up and different design.  Thus the results of this paper presents novel results of the subject area.

On page 5 row 202 the latter 7.0% should be removed.

Editorial minor notice: In the reference list the centered alignment of the text makes the word spaces too wide before active internet links